# Exploiting easy data in online optimization

**Amir Sani**     **Gergely Neu**     **Alessandro Lazaric**
SequeL team, INRIA Lille – Nord Europe, France
{amir.sani,gergely.neu,alessandro.lazaric}@inria.fr

## Abstract

We consider the problem of online optimization, where a learner chooses a deci-
sion from a given decision set and suffers some loss associated with the decision
and the state of the environment. The learner's objective is to minimize its cu-
mulative regret against the best fixed decision *in hindsight*. Over the past few
decades numerous variants have been considered, with many algorithms designed
to achieve sub-linear regret in the worst case. However, this level of robustness
comes at a cost. Proposed algorithms are often over-conservative, failing to adapt
to the *actual* complexity of the loss sequence which is often far from the worst
case. In this paper we introduce a general algorithm that, provided with a "safe"
learning algorithm and an opportunistic "benchmark", can effectively combine
good worst-case guarantees with much improved performance on "easy" data.
We derive general theoretical bounds on the regret of the proposed algorithm and
discuss its implementation in a wide range of applications, notably in the prob-
lem of learning with shifting experts (a recent COLT open problem). Finally, we
provide numerical simulations in the setting of prediction with expert advice with
comparisons to the state of the art.

## 1   Introduction

We consider a general class of online decision-making problems, where a *learner* sequentially de-
cides which actions to take from a given decision set and suffers some loss associated with the
decision and the state of the *environment*. The learner's goal is to minimize its cumulative loss as
the interaction between the learner and the environment is repeated. Performance is usually mea-
sured with regard to *regret*; that is, the difference between the cumulative loss of the algorithm and
the best single decision over the horizon in the decision set. The objective of the learning algorithm
is to guarantee that the per-round regret converges to zero as time progresses. This general setting
includes a wide range of applications such as online linear pattern recognition, sequential investment
and time series prediction.

Numerous variants of this problem were considered over the last few decades, mainly differing in the
shape of the decision set (see [6] for an overview). One of the most popular variants is the problem
of *prediction with expert advice*, where the decision set is the $N$-dimensional simplex and the per-
round losses are linear functions of the learner's decision. In this setting, a number of algorithms are
known to guarantee regret of order $\sqrt{T}$ after $T$ repetitions of the game. Another well-studied setting
is online convex optimization (OCO), where the decision set is a convex subset of $\mathbb{R}^d$ and the loss
functions are convex and smooth. Again, a number of simple algorithms are known to guarantee
a worst-case regret of order $\sqrt{T}$ in this setting. These results hold for any (possibly adversarial)
assignment of the loss sequences. Thus, these algorithms are guaranteed to achieve a decreasing
per-round regret that approaches the performance of the best fixed decision in hindsight even in the
worst case. Furthermore, these guarantees are unimprovable in the sense that there exist sequences
of loss functions where the learner suffers $\Omega(\sqrt{T})$ regret no matter what algorithm the learner uses.
However this robustness comes at a cost. These algorithms are often overconservative and fail to
adapt to the *actual* complexity of the loss sequence, which in practice is often far from the worst

possible. In fact, it is well known that making some assumptions on the loss generating mechanism improves the regret guarantees. For instance, the simple strategy of *following the leader* (FTL, otherwise known as *fictitious play* in game theory, see, e.g., [6, Chapter 7]), which at each round picks the single decision that minimizes the total losses so far, guarantees $\mathcal{O}(\log T)$ regret in the expert setting when assuming i.i.d. loss vectors. The same strategy also guarantees $\mathcal{O}(\log T)$ regret in the OCO setting, when assuming all loss functions are strongly convex. On the other hand, the risk of using this strategy is that it's known to suffer $\Omega(T)$ regret in the worst case.

This paper focuses on how to distinguish between "easy" and "hard" problem instances, while achieving the best possible guarantees on both types of loss sequences. This problem recently received much attention in a variety of settings (see, e.g., [8] and [13]), but most of the proposed solutions required the development of ad-hoc algorithms for each specific scenario and definition of "easy" problem. Another obvious downside of such ad-hoc solutions is that their theoretical analysis is often quite complicated and difficult to generalize to more complex problems. In the current paper, we set out to define an algorithm providing a general structure that can be instantiated in a wide range of settings by simply plugging in the most appropriate choice of two algorithms for learning on "easy" and "hard" problems.

Aside from exploiting easy data, our method has other potential applications. For example, in some sensitive applications we may want to protect ourselves from complete catastrophe, rather than take risks for higher payoffs. In fact, our work builds directly on the results of Even-Dar et al. [9], who point out that learning algorithms in the experts setting may fail to satisfy the rather natural requirement of performing *strictly better* than a trivial algorithm that merely decides on which expert to follow by uniform coin flips. While Even-Dar et al. propose methods that achieve this goal, they leave open an obvious open question. Is it possible to strictly improve the performance of an existing (and possibly naïve) solution by means of principled online learning methods? This problem can be seen as the polar opposite of failing to exploit easy data. In this paper, we push the idea of Even-Dar et al. one step further. We construct learning algorithms with order-optimal regret bounds, *while also guaranteeing that their cumulative loss is within a constant factor of some pre-defined strategy* referred to as the benchmark. We stress that this property is much stronger than simply guaranteeing $\mathcal{O}(1)$ regret with respect to some fixed distribution $D$ as done by Even-Dar et al. [9] since we allow comparisons to *any fixed strategy that is even allowed to learn*. Our method guarantees that replacing an existing solution can be done at a negligible price in terms of output performance with additional strong guarantees on the worst-case performance. However, in what follows, we will only regard this aspect of our results as an interesting consequence while emphasizing the ability of our algorithm to exploit easy data. Our general structure, referred to as $(\mathcal{A}, \mathcal{B})$-PROD, receives a learning algorithm $\mathcal{A}$ and a benchmark $\mathcal{B}$ as input. Depending on the online optimization setting, it is enough to set $\mathcal{A}$ to any learning algorithm with performance guarantees on "hard" problems and $\mathcal{B}$ to an opportunistic strategy exploiting the structure of "easy" problems. $(\mathcal{A}, \mathcal{B})$-PROD smoothly mixes the decisions of $\mathcal{A}$ and $\mathcal{B}$, achieving the best possible guarantees of both.

## 2 Online optimization with a benchmark

---

**Parameters**: set of decisions $\mathcal{S}$, number of rounds $T$;
**For all** $t = 1, 2, \ldots, T$, **repeat**

1. The environment chooses loss function $f_t : \mathcal{S} \to [0, 1]$.

2. The learner chooses a decision $x_t \in \mathcal{S}$.

3. The environment reveals $f_t$ (possibly chosen depending on the past history of losses and decisions).

4. The forecaster suffers loss $f_t(x_t)$.

---

Figure 1: The protocol of online optimization.

We now present the formal setting and an algorithm for online optimization with a benchmark. The interaction protocol between the learner and the environment is formally described on Figure 1. The online optimization problem is characterized by the decision set $\mathcal{S}$ and the class $\mathcal{F} \subseteq [0, 1]^{\mathcal{S}}$ of loss functions utilized by the environment. The performance of the learner is usually measured in terms of the *regret*, defined as $R_T = \sup_{x \in \mathcal{S}} \sum_{t=1}^{T} \big(f_t(x_t) - f_t(x)\big)$. We say that an algorithm *learns* if it makes decisions so that $R_T = o(T)$.

Let $\mathcal{A}$ and $\mathcal{B}$ be two online optimization algorithms that map observation histories to decisions in a possibly randomized fashion. For a formal definition, we fix a time index $t \in [T] = \{1, 2, \ldots, T\}$ and define the observation history (or in short, the history) at the end of round $t-1$ as $\mathcal{H}_{t-1} = (f_1, \ldots, f_{t-1})$. $\mathcal{H}_0$ is defined as the empty set. Furthermore, define the random variables $U_t$ and $V_t$, drawn from the standard uniform distribution, independently of $\mathcal{H}_{t-1}$ and each other. The learning algorithms $\mathcal{A}$ and $\mathcal{B}$ are formally defined as mappings from $\mathcal{F}^* \times [0, 1]$ to $\mathcal{S}$ with their respective decisions given as

$$a_t \stackrel{\text{def}}{=} \mathcal{A}(\mathcal{H}_{t-1}, U_t) \qquad \text{and} \qquad b_t \stackrel{\text{def}}{=} \mathcal{B}(\mathcal{H}_{t-1}, V_t).$$

Finally, we define a *hedging strategy* $\mathcal{C}$ that produces a decision $x_t$ based on the history of decisions proposed by $\mathcal{A}$ and $\mathcal{B}$, with the possible help of some external randomness represented by the uniform random variable $W_t$ as $x_t = \mathcal{C}\left(a_t, b_t, \mathcal{H}^*_{t-1}, W_t\right)$. Here, $\mathcal{H}^*_{t-1}$ is the simplified history consisting of $\left(f_1(a_1), f_1(b_1), \ldots, f_{t-1}(a_{t-1}), f_{t-1}(b_{t-1})\right)$ and $\mathcal{C}$ bases its decisions only on the past losses incurred by $\mathcal{A}$ and $\mathcal{B}$ without using any further information on the loss functions. The total expected loss of $\mathcal{C}$ is defined as $\widehat{L}_T(\mathcal{C}) = \mathbb{E}[\sum_{t=1}^{T} f_t(x_t)]$, where the expectation integrates over the possible realizations of the internal randomization of $\mathcal{A}, \mathcal{B}$ and $\mathcal{C}$. The total expected losses of $\mathcal{A}, \mathcal{B}$ and any fixed decision $x \in \mathcal{S}$ are similarly defined.

Our goal is to define a hedging strategy with low regret against a benchmark strategy $\mathcal{B}$, while also enjoying near-optimal guarantees on the worst-case regret against the best decision in hindsight. The (expected) regret of $\mathcal{C}$ against any fixed decision $x \in \mathcal{S}$ and against the benchmark, are defined as

$$\mathfrak{R}_T(\mathcal{C}, x) = \mathbb{E}\left[\sum_{t=1}^{T}\left(f_t(x_t) - f_t(x)\right)\right], \quad \mathfrak{R}_T(\mathcal{C}, \mathcal{B}) = \mathbb{E}\left[\sum_{t=1}^{T}\left(f_t(x_t) - f_t(b_t)\right)\right].$$

Our hedging strategy, $(\mathcal{A}, \mathcal{B})$-PROD, is based on the classic PROD algorithm popularized by Cesa-Bianchi et al. [7] and builds on a variant of PROD called $D$-PROD, proposed in Even-Dar et al. [9], which (when properly tuned) achieves constant regret against the performance of a fixed distribution $D$ over experts, while guaranteeing $\mathcal{O}(\sqrt{T \log T})$ regret against the best expert in hindsight. Our variant $(\mathcal{A}, \mathcal{B})$-PROD (shown in Figure 2) is based on the observation that it is not necessary to use a fixed distribution $D$ in the definition of the benchmark, but actually any learning algorithm or signal can be used as a baseline. $(\mathcal{A}, \mathcal{B})$-PROD maintains two weights, balancing the advice of learning algorithm $\mathcal{A}$ and a benchmark $\mathcal{B}$. The benchmark weight is defined as $w_{1,\mathcal{B}} \in (0, 1)$ and is kept unchanged during the entire learning process. The initial weight assigned to $\mathcal{A}$ is $w_{1,\mathcal{A}} = 1 - w_{1,\mathcal{B}}$, and in the remaining rounds $t = 2, 3, \ldots, T$ is updated as

---

**Input**: learning rate $\eta \in (0, 1/2]$, initial weights $\{w_{1,\mathcal{A}}, w_{1,\mathcal{B}}\}$, num. of rounds $T$;
**For all** $t = 1, 2, \ldots, T$, **repeat**

1. Let
$$s_t = \frac{w_{t,\mathcal{A}}}{w_{t,\mathcal{A}} + w_{1,\mathcal{B}}}.$$

2. Observe $a_t$ and $b_t$ and predict
$$x_t = \begin{cases} a_t & \text{with probability } s_t, \\ b_t & \text{otherwise.} \end{cases}$$

3. Observe $f_t$ and suffer loss $f_t(x_t)$.

4. Feed $f_t$ to $\mathcal{A}$ and $\mathcal{B}$.

5. Compute $\delta_t = f_t(b_t) - f_t(a_t)$ and set
$$w_{t+1,\mathcal{A}} = w_{t,\mathcal{A}} \cdot (1 + \eta \delta_t).$$

Figure 2: $(\mathcal{A}, \mathcal{B})$-PROD

---

$$w_{t,\mathcal{A}} = w_{1,\mathcal{A}} \prod_{s=1}^{t-1}\left(1 - \eta\left(f_s(a_s) - f_s(b_s)\right)\right),$$

where the difference between the losses of $\mathcal{A}$ and $\mathcal{B}$ is used. Output $x_t$ is set to $a_t$ with probability $s_t = w_{t,\mathcal{A}}/(w_{t,\mathcal{A}} + w_{1,\mathcal{B}})$, otherwise it is set to $b_t$.[1] The following theorem states the performance guarantees for $(\mathcal{A}, \mathcal{B})$-PROD.

**Theorem 1** (cf. Lemma 1 in [9]). *For any assignment of the loss sequence, the total expected loss of* $(\mathcal{A}, \mathcal{B})$-PROD *initialized with weights* $w_{1,\mathcal{B}} \in (0, 1)$ *and* $w_{1,\mathcal{B}} = 1 - w_{1,\mathcal{A}}$ *simultaneously satisfies*

$$\widehat{L}_T\left((\mathcal{A}, \mathcal{B})\text{-PROD}\right) \leq \widehat{L}_T(\mathcal{A}) + \eta \sum_{t=1}^{T}\left(f_t(b_t) - f_t(a_t)\right)^2 - \frac{\log w_{1,\mathcal{A}}}{\eta}$$

*and*

$$\widehat{L}_T\left((\mathcal{A}, \mathcal{B})\text{-PROD}\right) \leq \widehat{L}_T(\mathcal{B}) - \frac{\log w_{1,\mathcal{B}}}{\eta}.$$

The proof directly follows from the PROD analysis of Cesa-Bianchi et al. [7]. Next, we suggest a parameter setting for $(\mathcal{A}, \mathcal{B})$-PROD that guarantees constant regret against the benchmark $\mathcal{B}$ and $\mathcal{O}(\sqrt{T \log T})$ regret against the learning algorithm $\mathcal{A}$ in the worst case.

**Corollary 1.** *Let $C \geq 1$ be an upper bound on the total benchmark loss $\widehat{L}_T(\mathcal{B})$. Then setting $\eta = 1/2 \cdot \sqrt{(\log C)/C} < 1/2$ and $w_{1,\mathcal{B}} = 1 - w_{1,\mathcal{A}} = 1 - \eta$ simultaneously guarantees*

$$\mathfrak{R}_T\big((\mathcal{A}, \mathcal{B})\text{-PROD}, x\big) \leq \mathfrak{R}_T(\mathcal{A}, x) + 2\sqrt{C \log C}$$

*for any $x \in \mathcal{S}$ and*

$$\mathfrak{R}_T\big((\mathcal{A}, \mathcal{B})\text{-PROD}, \mathcal{B}\big) \leq 2 \log 2$$

*against any assignment of the loss sequence.*

Notice that for any $x \in \mathcal{S}$, the previous bounds can be written as

$$\mathfrak{R}_T((\mathcal{A}, \mathcal{B})\text{-PROD}, x) \leq \min\left\{\mathfrak{R}_T(\mathcal{A}, x) + 2\sqrt{C \log C}, \mathfrak{R}_T(\mathcal{B}, x) + 2 \log 2\right\},$$

which states that $(\mathcal{A}, \mathcal{B})$-PROD achieves the minimum between the regret of the benchmark $\mathcal{B}$ and learning algorithm $\mathcal{A}$ plus an additional regret of $\mathcal{O}(\sqrt{C \log C})$. If we consider that in most online optimization settings, the worst-case regret for a learning algorithm is $\mathcal{O}(\sqrt{T})$, the previous bound shows that at the cost of an additional factor of $\mathcal{O}(\sqrt{T \log T})$ in the worst case, $(\mathcal{A}, \mathcal{B})$-PROD performs as well as the benchmark, which is very useful whenever $\mathfrak{R}_T(\mathcal{B}, x)$ is small. This suggests that if we set $\mathcal{A}$ to a learning algorithm with worst-case guarantees on "difficult" problems and $\mathcal{B}$ to an algorithm with very good performance only on "easy" problems, then $(\mathcal{A}, \mathcal{B})$-PROD successfully adapts to the difficulty of the problem by finding a suitable mixture of $\mathcal{A}$ and $\mathcal{B}$. Furthermore, as discussed by Even-Dar et al. [9], we note that in this case the PROD update rule is crucial to achieve this result: any algorithm that bases its decisions solely on the *cumulative difference* between $f_t(a_t)$ and $f_t(b_t)$ is bound to suffer an additional regret of $\mathcal{O}(\sqrt{T})$ on both $\mathcal{A}$ and $\mathcal{B}$. While HEDGE and follow-the-perturbed-leader (FPL) both fall into this category, it can be easily seen that this is not the case for PROD. A similar observation has been made by de Rooij et al. [8], who discuss the possibility of combining a robust learning algorithm and FTL by HEDGE and conclude that this approach is insufficient for their goals – see also Sect. 3.1.

Finally, we note that the parameter proposed in Corollary 1 can hardly be computed in practice, since an upper-bound on the loss of the benchmark $\widehat{L}_T(\mathcal{B})$ is rarely available. Fortunately, we can adapt an improved version of PROD with adaptive learning rates recently proposed by Gaillard et al. [11] and obtain an anytime version of $(\mathcal{A}, \mathcal{B})$-PROD. The resulting algorithm and its corresponding bounds are reported in App. B.

## 3 Applications

The following sections apply our results to special cases of online optimization. Unless otherwise noted, all theorems are direct consequences of Corollary 1 and thus their proofs are omitted.

### 3.1 Prediction with expert advice

We first consider the most basic online optimization problem of prediction with expert advice. Here, $\mathcal{S}$ is the $N$-dimensional simplex $\Delta_N = \left\{x \in \mathbb{R}_+^N : \sum_{i=1}^N x_i = 1\right\}$ and the loss functions are linear, that is, the loss of any decision $x \in \Delta_N$ in round $t$ is given as the inner product $f_t(x) = x^\top \ell_t$ and $\ell_t \in [0, 1]^N$ is the loss vector in round $t$. Accordingly, the family $\mathcal{F}$ of loss functions can be equivalently represented by the set $[0, 1]^N$. Many algorithms are known to achieve the optimal regret guarantee of $\mathcal{O}(\sqrt{T \log N})$ in this setting, including HEDGE (so dubbed by Freund and Schapire [10], see also the seminal works of Littlestone and Warmuth [20] and Vovk [23]) and the follow-the-perturbed-leader (FPL) prediction method of Hannan [16], later rediscovered by Kalai and Vempala [19]. However, as de Rooij et al. [8] note, these algorithms are usually too conservative to exploit "easily learnable" loss sequences and might be significantly outperformed by a simple strategy known as follow-the-leader (FTL), which predicts $b_t = \arg\min_{x \in \mathcal{S}} x^\top \sum_{s=1}^{t-1} \ell_s$. For instance, FTL is known to be optimal in the case of i.i.d. losses, where it achieves a regret of $\mathcal{O}(\log T)$. As a direct consequence of Corollary 1, we can use the general structure of $(\mathcal{A}, \mathcal{B})$-PROD to match the performance of FTL on easy data, and at the same time, obtain the same worst-case guarantees of standard algorithms for prediction with expert advice. In particular, if we set FTL as the benchmark $\mathcal{B}$ and ADAHEDGE (see [8]) as the learning algorithm $\mathcal{A}$, we obtain the following.

**Theorem 2.** *Let $\mathcal{S} = \Delta_N$ and $\mathcal{F} = [0,1]^N$. Running $(\mathcal{A}, \mathcal{B})$-PROD with $\mathcal{A} = $ ADAHEDGE and $\mathcal{B} = $ FTL, with the parameter setting suggested in Corollary 1 simultaneously guarantees*

$$\mathfrak{R}_T\big((\mathcal{A}, \mathcal{B})\text{-PROD}, x\big) \leq \mathfrak{R}_T(\text{ADAHEDGE}, x) + 2\sqrt{C \log C} \leq \sqrt{\frac{L_T^*(T - L_T^*)}{T} \log N} + 2\sqrt{C \log C}$$

*for any $x \in \mathcal{S}$, where $L_T^* = \min_{x \in \Delta_N} L_T(x)$, and*

$$\mathfrak{R}_T\big((\mathcal{A}, \mathcal{B})\text{-PROD}, \text{FTL}\big) \leq 2 \log 2.$$

*against any assignment of the loss sequence.*

While we recover the worst-case guarantee of $\mathcal{O}(\sqrt{T \log N})$ plus an additional regret $\mathcal{O}(\sqrt{T \log T})$ on "hard" loss sequences, on "easy" problems we inherit the good performance of FTL.

**Comparison with FLIPFLOP.** The FLIPFLOP algorithm proposed by de Rooij et al. [8] addresses the problem of constructing algorithms that perform nearly as well as FTL on easy problems while retaining optimal guarantees on all possible loss sequences. More precisely, FLIPFLOP is a HEDGE algorithm where the learning rate $\eta$ alternates between infinity (corresponding to FTL) and the value suggested by ADAHEDGE depending on the cumulative mixability gaps over the two regimes. The resulting algorithm is guaranteed to achieve the regret guarantees of

$$\mathfrak{R}_T(\text{FLIPFLOP}, x) \leq 5.64 \mathfrak{R}_T(\text{FTL}, x) + 3.73$$

and

$$\mathfrak{R}_T(\text{FLIPFLOP}, x) \leq 5.64 \sqrt{\frac{L_T^*(T - L_T^*)}{T} \log N} + \mathcal{O}(\log N)$$

against any fixed $x \in \Delta_N$ at the same time. Notice that while the guarantees in Thm. 2 are very similar in nature to those of de Rooij et al. [8] concerning FLIPFLOP, the two results are slightly different. The first difference is that our worst-case bounds are inferior to theirs by a factor of order $\sqrt{T \log T}$.[2] On the positive side, our guarantees are much stronger when FTL outperforms ADAHEDGE. To see this, observe that their regret bound can be rewritten as

$$L_T(\text{FLIPFLOP}) \leq L_T(\text{FTL}) + 4.64\big(L_T(\text{FTL}) - \inf_x L_T(x)\big) + 3.73,$$

whereas our result replaces the last two terms by $2 \log 2$.[3] The other advantage of our result is that we can directly bound the *total loss* of our algorithm in terms of the *total loss of* ADAHEDGE (see Thm. 1). This is to be contrasted with the result of de Rooij et al. [8], who upper bound their *regret* in terms of the *regret bound* of ADAHEDGE, which may not be tight and be much worse in practice than the actual performance of ADAHEDGE. All these advantages of our approach stem from the fact that we smoothly mix the predictions of ADAHEDGE and FTL, while FLIPFLOP explicitly follows one policy or the other for extended periods of time, potentially accumulating unnecessary losses when switching too late or too early. Finally, we note that as FLIPFLOP is a sophisticated algorithm specifically designed for balancing the performance of ADAHEDGE and FTL in the expert setting, we cannot reasonably hope to beat its performance in every respect by using our general-purpose algorithm. Notice however that the analysis of FLIPFLOP is difficult to generalize to other learning settings such as the ones we discuss in the sections below.

**Comparison with $D$-PROD.** In the expert setting, we can also use a straightforward modification of the $D$-PROD algorithm originally proposed by Even-Dar et al. [9]: This variant of PROD includes the benchmark $\mathcal{B}$ in $\Delta_N$ as an additional expert and performs PROD updates for each base expert using the difference between the expert and benchmark losses. While the worst-case regret of this algorithm is of $\mathcal{O}(\sqrt{C \log C \log N})$, which is asymptotically inferior to the guarantees given by Thm. 2, $D$-PROD also has its merits in some special cases. For instance, in a situation where the total loss of FTL and the regret of ADAHEDGE are both $\Theta(\sqrt{T})$, $D$-PROD guarantees a regret of $\mathcal{O}(T^{1/4})$ while the $(\mathcal{A}, \mathcal{B})$-PROD guarantee remains $\mathcal{O}(\sqrt{T})$.

## 3.2 Tracking the best expert

We now turn to the problem of tracking the best expert, where the goal of the learner is to control the regret against the best fixed strategy that is allowed to change its prediction at most $K$ times during the entire decision process (see, e.g., [18, 14]). The regret of an algorithm $\mathcal{A}$ producing predictions $a_1, \ldots, a_T$ against an arbitrary sequence of decisions $y_{1:T} \in \mathcal{S}^T$ is defined as

$$\mathfrak{R}_T(\mathcal{A}, y_{1:T}) = \sum_{t=1}^{T} \big(f_t(a_t) - f_t(y_t)\big).$$

Regret bounds in this setting typically depend on the complexity of the sequence $y_{1:T}$ as measured by the number decision switches $C(y_{1:T}) = \{t \in \{2, \ldots, T\} : y_t \neq y_{t-1}\}$. For example, a properly tuned version of the FIXED-SHARE (FS) algorithm of Herbster and Warmuth [18] guarantees that $\mathfrak{R}_T(\mathrm{FS}, y_{1:T}) = \mathcal{O}\big(C(y_{1:T})\sqrt{T \log N}\big)$. This upper bound can be tightened to $\mathcal{O}(\sqrt{KT \log N})$ when the learner knows an upper bound $K$ on the complexity of $y_{1:T}$. While this bound is unimprovable in general, one might wonder if it is possible to achieve better performance when the loss sequence is easy. This precise question was posed very recently as a COLT open problem by Warmuth and Koolen [24]. The generality of our approach allows us to solve their open problem by using $(\mathcal{A}, \mathcal{B})$-PROD as a master algorithm to combine an opportunistic strategy with a principled learning algorithm. The following theorem states the performance of the $(\mathcal{A}, \mathcal{B})$-PROD-based algorithm.

**Theorem 3.** *Let $\mathcal{S} = \Delta_N$, $\mathcal{F} = [0,1]^N$ and $y_{1:T}$ be any sequence in $\mathcal{S}$ with known complexity $K = C(y_{1:T})$. Running $(\mathcal{A}, \mathcal{B})$-PROD with an appropriately tuned instance of $\mathcal{A} = \mathrm{FS}$ (see [18]), with the parameter setting suggested in Corollary 1 simultaneously guarantees*

$$\mathfrak{R}_T\big((\mathcal{A}, \mathcal{B})\text{-PROD}, y_{1:T}\big) \leq \mathfrak{R}_T(\mathrm{FS}, y_{1:T}) + 2\sqrt{C \log C} = \mathcal{O}(\sqrt{KT \log N}) + 2\sqrt{C \log C}$$

*for any $x \in \mathcal{S}$ and*

$$\mathfrak{R}_T\big((\mathcal{A}, \mathcal{B})\text{-PROD}, \mathcal{B}\big) \leq 2 \log 2.$$

*against any assignment of the loss sequence.*

The remaining problem is then to find a benchmark that works well on "easy" problems, notably when the losses are i.i.d. in $K$ (unknown) segments of the rounds $1, \ldots, T$. Out of the strategies suggested by Warmuth and Koolen [24], we analyze a windowed variant of FTL (referred to as FTL($w$)) that bases its decision at time $t$ on losses observed in the time window $[t-w-1, t-1]$ and picks expert $b_t = \arg\min_{x \in \Delta_N} x^\top \sum_{s=t-w-1}^{t-1} \ell_s$. The next proposition (proved in the appendix) gives a performance guarantee for FTL($w$) with an optimal parameter setting.

**Proposition 1.** *Assume that there exists a partition of $[1, T]$ into $K$ intervals such that the losses are generated i.i.d. within each interval. Furthermore, assume that the expectation of the loss of the best expert within each interval is at least $\delta$ away from the expected loss of all other experts. Then, setting $w = \lceil 4 \log(NT/K)/\delta^2 \rceil$, the regret of FTL($w$) is upper bounded for any $y_{1:T}$ as*

$$\mathbb{E}\big[\mathfrak{R}_T(\mathrm{FTL}(w), y_{1:T})\big] \leq \frac{4K}{\delta^2} \log(NT/K) + 2K,$$

*where the expectation is taken with respect to the distribution of the losses.*

## 3.3 Online convex optimization

Here we consider the problem of online convex optimization (OCO), where $\mathcal{S}$ is a convex and closed subset of $\mathbb{R}^d$ and $\mathcal{F}$ is the family of convex functions on $\mathcal{S}$. In this setting, if we assume that the loss functions are smooth (see [25]), an appropriately tuned version of the online gradient descent (OGD) is known to achieve a regret of $\mathcal{O}(\sqrt{T})$. As shown by Hazan et al. [17], if we additionally assume that the environment plays *strongly convex* loss functions and tune the parameters of the algorithm accordingly, the same algorithm can be used to guarantee an improved regret of $\mathcal{O}(\log T)$. Furthermore, they also show that FTL enjoys essentially the same guarantees. The question whether the two guarantees can be combined was studied by Bartlett et al. [4], who present the adaptive online gradient descent (AOGD) algorithm that guarantees $\mathcal{O}(\log T)$ regret when the aggregated loss functions $F_t = \sum_{s=1}^{t} f_s$ are strongly convex for all $t$, while retaining the $\mathcal{O}(\sqrt{T})$ bounds if this is not the case. The next theorem shows that we can replace their complicated analysis by our general argument and show essentially the same guarantees.

**Theorem 4.** *Let $\mathcal{S}$ be a convex closed subset of $\mathbb{R}^d$ and $\mathcal{F}$ be the family of smooth convex functions on $\mathcal{S}$. Running $(\mathcal{A}, \mathcal{B})$-PROD with an appropriately tuned instance of $\mathcal{A} = $ OGD (see [25]) and $\mathcal{B} = $ FTL, with the parameter setting suggested in Corollary 1 simultaneously guarantees*

$$\mathfrak{R}_T\big((\mathcal{A}, \mathcal{B})\text{-PROD}, x\big) \leq \mathfrak{R}_T(\text{OGD}, x) + 2\sqrt{C \log C} = \mathcal{O}(\sqrt{T}) + 2\sqrt{C \log C}$$

*for any $x \in \mathcal{S}$ and*

$$\mathfrak{R}_T\big((\mathcal{A}, \mathcal{B})\text{-PROD}, \text{FTL}\big) \leq 2 \log 2.$$

*against any assignment of the loss sequence. In particular, this implies that*

$$\mathfrak{R}_T\big((\mathcal{A}, \mathcal{B})\text{-PROD}, x\big) = \mathcal{O}(\log T)$$

*if the loss functions are strongly convex.*

Similar to the previous settings, at the cost of an additional regret of $\mathcal{O}(\sqrt{T \log T})$ in the worst case, $(\mathcal{A}, \mathcal{B})$-PROD successfully adapts to the "easy" loss sequences, which in this case corresponds to strongly convex functions, on which it achieves a $\mathcal{O}(\log T)$ regret.

### 3.4 Learning with two-points-bandit feedback

We consider the multi-armed bandit problem with two-point feedback, where we assume that in each round $t$, the learner picks one arm $I_t$ in the decision set $\mathcal{S} = \{1, 2, \ldots, K\}$ and *also has the possibility to choose and observe the loss of another arm $J_t$*. The learner suffers the loss $f_t(I_t)$. Unlike the settings considered in the previous sections, the learner only gets to observe the loss function for arms $I_t$ and $J_t$. This is a special case of the partial-information game recently studied by Seldin et al. [21]. A similar model has also been studied as a simplified version of online convex optimization with partial feedback [1]. While this setting does not entirely conform to our assumptions concerning $\mathcal{A}$ and $\mathcal{B}$, observe that a hedging strategy $\mathcal{C}$ defined over $\mathcal{A}$ and $\mathcal{B}$ only requires access to *the losses suffered by the two algorithms and not the entire loss functions*. Formally, we give $\mathcal{A}$ and $\mathcal{B}$ access to the decision set $\mathcal{S}$, and $\mathcal{C}$ to $\mathcal{S}^2$. The hedging strategy $\mathcal{C}$ selects the pair $(I_t, J_t)$ based on the arms suggested by $\mathcal{A}$ and $\mathcal{B}$ as:

$$(I_t, J_t) = \begin{cases} (a_t, b_t) & \text{with probability } s_t, \\ (b_t, a_t) & \text{with probability } 1 - s_t. \end{cases}$$

The probability $s_t$ is a well-defined deterministic function of $\mathcal{H}_{t-1}^*$, thus the regret bound of $(\mathcal{A}, \mathcal{B})$-PROD can be directly applied. In this case, "easy" problems correspond to i.i.d. loss sequences (with a fixed gap between the expected losses), for which the UCB algorithm of Auer et al. [2] is guaranteed to have a $\mathcal{O}(\log T)$ regret, while on "hard" problems, we can rely on the EXP3 algorithm of Auer et al. [3] which suffers a regret of $\mathcal{O}(\sqrt{TK})$ in the worst case. The next theorem gives the performance guarantee of $(\mathcal{A}, \mathcal{B})$-PROD when combining UCB and EXP3.

**Theorem 5.** *Consider the multi-armed bandit problem with $K$ arms and two-point feedback. Running $(\mathcal{A}, \mathcal{B})$-PROD with an appropriately tuned instance of $\mathcal{A} = $ EXP3 (see [3]) and $\mathcal{B} = $ UCB (see [2]), with the parameter setting suggested in Corollary 1 simultaneously guarantees*

$$\mathfrak{R}_T\big((\mathcal{A}, \mathcal{B})\text{-PROD}, x\big) \leq \mathfrak{R}_T(\text{EXP3}, x) + 2\sqrt{C \log C} = \mathcal{O}(\sqrt{TK \log K}) + 2\sqrt{C \log C}$$

*for any arm $x \in \{1, 2, \ldots, K\}$ and*

$$\mathfrak{R}_T\big((\mathcal{A}, \mathcal{B})\text{-PROD}, \text{UCB}\big) \leq 2 \log 2.$$

*against any assignment of the loss sequence. In particular, if the losses are generated in an i.i.d. fashion and there exists a unique best arm $x^* \in \mathcal{S}$, then*

$$\mathbb{E}\big[\mathfrak{R}_T\big((\mathcal{A}, \mathcal{B})\text{-PROD}, x\big)\big] = \mathcal{O}(\log T),$$

*where the expectation is taken with respect to the distribution of the losses.*

This result shows that even in the multi-armed bandit setting, we can achieve nearly the best performance in both "hard" and "easy" problems given that we are allowed to pull two arms at the time. This result is to be contrasted with those of Bubeck and Slivkins [5], later improved by Seldin and Slivkins [22], who consider the standard one-point feedback setting. The algorithm of Seldin and Slivkins, called EXP3++ is a variant of the EXP3 algorithm that simultaneously guarantees $\mathcal{O}(\log^2 T)$ regret in stochastic environments while retaining the regret bound of $\mathcal{O}(\sqrt{TK \log K})$ in the adversarial setting. While our result holds under stronger assumptions, Thm. 5 shows that $(\mathcal{A}, \mathcal{B})$-PROD is not restricted to work only in full-information settings. Once again, we note that such a result cannot be obtained by simply combining the predictions of UCB and EXP3 by a generic learning algorithm as HEDGE.

# 4 Empirical Results

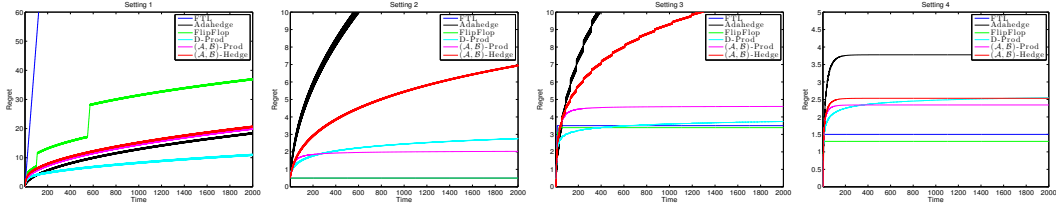

Figure 3: Hand tuned loss sequences from de Rooij et al. [8]

We study the performance of $(\mathcal{A}, \mathcal{B})$-PROD in the experts setting to verify the theoretical results of Thm. 2, show the importance of the $(\mathcal{A}, \mathcal{B})$-PROD weight update rule and compare to FLIPFLOP. We report the performance of FTL, ADAHEDGE, FLIPFLOP, and $\mathcal{B} = $ FTL and $\mathcal{A} = $ ADAHEDGE for the anytime versions of $D$-PROD, $(\mathcal{A}, \mathcal{B})$-PROD, and $(\mathcal{A}, \mathcal{B})$-HEDGE, a variant of $(\mathcal{A}, \mathcal{B})$-PROD where an exponential weighting scheme is used. We consider the two-expert settings defined by de Rooij et al. [8] where deterministic loss sequences of $T = 2000$ steps are designed to obtain different configurations. (We refer to [8] for a detailed specification of the settings.) The results are reported in Figure 3. The first remark is that the performance of $(\mathcal{A}, \mathcal{B})$-PROD is always comparable with the best algorithm between $\mathcal{A}$ and $\mathcal{B}$. In setting 1, although FTL suffers linear regret, $(\mathcal{A}, \mathcal{B})$-PROD rapidly adjusts the weights towards ADAHEDGE and finally achieves the same order of performance. In settings 2 and 3, the situation is reversed since FTL has a constant regret, while ADAHEDGE has a regret of order of $\sqrt{T}$. In this case, after a short initial phase where $(\mathcal{A}, \mathcal{B})$-PROD has an increasing regret, it stabilizes on the same performance as FTL. In setting 4 both ADAHEDGE and FTL have a constant regret and $(\mathcal{A}, \mathcal{B})$-PROD attains the same performance. These results match the behavior predicted in the bound of Thm. 2, which guarantees that the regret of $(\mathcal{A}, \mathcal{B})$-PROD is roughly the minimum of FTL and ADAHEDGE. As discussed in Sect. 2, the PROD update rule used in $(\mathcal{A}, \mathcal{B})$-PROD plays a crucial role to obtain a constant regret against the benchmark, while other rules, such as the exponential update used in $(\mathcal{A}, \mathcal{B})$-HEDGE, may fail in finding a suitable mix between $\mathcal{A}$ and $\mathcal{B}$. As illustrated in settings 2 and 3, $(\mathcal{A}, \mathcal{B})$-HEDGE suffers a regret similar to ADAHEDGE and it fails to take advantage of the good performance of FTL, which has a constant regret. In setting 1, $(\mathcal{A}, \mathcal{B})$-HEDGE performs as well as $(\mathcal{A}, \mathcal{B})$-PROD because FTL is constantly worse than ADAHEDGE and its corresponding weight is decreased very quickly, while in setting 4 both FTL and ADAHEDGE achieves a constant regret and so does $(\mathcal{A}, \mathcal{B})$-HEDGE. Finally, we compare $(\mathcal{A}, \mathcal{B})$-PROD and FLIPFLOP. As discussed in Sect. 2, the two algorithms share similar theoretical guarantees with potential advantages of one on the other depending on the specific setting. In particular, FLIPFLOP performs slightly better in settings 2, 3, and 4, whereas $(\mathcal{A}, \mathcal{B})$-PROD obtains smaller regret in setting 1, where the constants in the FLIPFLOP bound show their teeth. While it is not possible to clearly rank the two algorithms, $(\mathcal{A}, \mathcal{B})$-PROD clearly avoids the pathological behavior exhibited by FLIPFLOP in setting 1. Finally, we note that the anytime version of $D$-PROD is slightly better than $(\mathcal{A}, \mathcal{B})$-PROD, but no consistent difference is observed.

# 5 Conclusions

We introduced $(\mathcal{A}, \mathcal{B})$-PROD, a general-purpose algorithm which receives a learning algorithm $\mathcal{A}$ and a benchmark strategy $\mathcal{B}$ as inputs and guarantees the best regret between the two. We showed that whenever $\mathcal{A}$ is a learning algorithm with worst-case performance guarantees and $\mathcal{B}$ is an opportunistic strategy exploiting a specific structure within the loss sequence, we obtain an algorithm which smoothly adapts to "easy" and "hard" problems. We applied this principle to a number of different settings of online optimization, matching the performance of existing ad-hoc solutions (e.g., AOGD in convex optimization) and solving the open problem of learning on "easy" loss sequences in the tracking the best expert setting proposed by Warmuth and Koolen [24]. We point out that the general structure of $(\mathcal{A}, \mathcal{B})$-PROD could be instantiated in many other settings and scenarios in online optimization, such as learning with switching costs [12, 15], and, more generally, in any problem where the objective is to improve over a given benchmark strategy. The main open problem is the extension of our techniques to work with one-point bandit feedback.

**Acknowledgements**   This work was supported by the French Ministry of Higher Education and Research and by the European Community's Seventh Framework Programme (FP7/2007-2013) under grant agreement 270327 (project CompLACS), and by FUI project Hermès.

## Footnotes

[1] For convex decision sets $\mathcal{S}$ and loss families $\mathcal{F}$, one can directly set $x_t = s_t a_t + (1 - s_t)b_t$ at no expense.

[2]In fact, the worst case for our bound is realized when $C = \Omega(T)$, which is precisely the case when ADAHEDGE has excellent performance as it will be seen in Sect. 4.

[3]While one can parametrize FLIPFLOP so as to decrease the gap between these bounds, the bound on $L_T(\text{FLIPFLOP})$ is always going to be linear in $\mathfrak{R}_T(\text{FLIPFLOP}, x)$.

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
