[Supplementary Material · benchmarks_supplement.pdf]

# A Proof of Corollary 1

The second part follows from the fact that $\log(1 - \eta)/\eta$ is an decreasing function on $\eta \in (0, 1/2)$. For the first part, we study two cases. In the first case, we assume that $\widehat{L}_T(\mathcal{B}) \leq \widehat{L}_T(\mathcal{A})$ holds, which proves the statement for this case. For the second case, we assume the contrary and notice that

$$\sum_{t=1}^{T}\big(f_t(b_t) - f_t(a_t)\big)^2 \leq \sum_{t=1}^{T}\big|f_t(b_t) - f_t(a_t)\big|$$

$$= \sum_{t=1}^{T}\big(f_t(b_t) - f_t(a_t)\big)^+ + \sum_{t=1}^{T}\big(f_t(b_t) - f_t(a_t)\big)^-,$$

where $(z)^+$ and $(z)^-$ are the positive and negative parts of $z \in \mathbb{R}$, respectively. Now observe that

$$\sum_{t=1}^{T}\big(f_t(b_t) - f_t(a_t)\big)^+ - \sum_{t=1}^{T}\big(f_t(b_t) - f_t(a_t)\big)^- = \widehat{L}_T(\mathcal{B}) - \widehat{L}_T(\mathcal{A}) \geq 0,$$

implying

$$\sum_{t=1}^{T}\big(f_t(b_t) - f_t(a_t)\big)^- \leq \sum_{t=1}^{T}\big(f_t(b_t) - f_t(a_t)\big)^+$$

and thus

$$\sum_{t=1}^{T}\big(f_t(b_t) - f_t(a_t)\big)^2 \leq 2\sum_{t=1}^{T}\big(f_t(b_t) - f_t(a_t)\big)^+ \leq 2\widehat{L}_T(\mathcal{B}) \leq 2C.$$

Plugging this result into the first bound of Thm. 1 and substituting the choice of $\eta$ gives the result.

# B Anytime $(\mathcal{A}, \mathcal{B})$-PROD

---
**Algorithm 1** Anytime $(\mathcal{A}, \mathcal{B})$-PROD

---
**Initialization**: $\eta_1 = 1/2$, $w_{1,\mathcal{A}} = w_{1,\mathcal{B}} = 1/2$
**For all** $t = 1, 2, \ldots, T$, **repeat**

1. Let

$$\eta_t = \sqrt{\frac{1}{1 + \sum_{s=1}^{t-1}(f_s(b_s) - f_s(a_s))^2}}$$

and

$$s_t = \frac{\eta_t w_{t,\mathcal{A}}}{\eta_t w_{t,\mathcal{A}} + w_{1,\mathcal{B}}/2}.$$

2. Observe $a_t$ and $b_t$ and predict

$$x_t = \begin{cases} a_t & \text{with probability } s_t, \\ b_t & \text{with probability } 1 - s_t. \end{cases}$$

3. Observe $f_t$ and suffer loss $f_t(x_t)$.
4. Feed $f_t$ to $\mathcal{A}$ and $\mathcal{B}$.
5. Compute $\delta_t = f_t(b_t) - f_t(a_t)$ and set

$$w_{t+1,\mathcal{A}} = w_{t,\mathcal{A}} \cdot (1 + \eta_{t-1}\delta_t)^{\eta_t/\eta_{t-1}}.$$

---

Algorithm 1 presents the adaptation of the adaptive-learning-rate PROD variant recently proposed by Gaillard et al. [11] to our setting. Following their analysis, we can prove the following performance guarantee concerning the adaptive version of $(\mathcal{A}, \mathcal{B})$-PROD.

**Theorem 6.** *Let $C$ be an upper bound on the total benchmark loss $\widehat{L}_T(\mathcal{B})$. Then anytime $(\mathcal{A},\mathcal{B})$-*
PROD *simultaneously guarantees*

$$\mathfrak{R}_T((\mathcal{A},\mathcal{B})\text{-}\text{PROD},x) \leq \mathfrak{R}_T(\mathcal{A},x) + K_T\sqrt{C+1} + 2K_T$$

*for any $x \in \mathcal{S}$ and*

$$\mathfrak{R}_T((\mathcal{A},\mathcal{B})\text{-}\text{PROD},\mathcal{B}) \leq 2\log 2 + 2K_T$$

*against any assignment of the loss sequence, where $K_T = \mathcal{O}(\log\log T)$.*

There are some notable differences between the guarantees given by the above theorem and Thm. 1. The most important difference is that the current statement guarantees an improved regret of $\mathcal{O}(\sqrt{T}\log\log T)$ instead of $\sqrt{T\log T}$ in the worst case – however, this comes at the price of an $\mathcal{O}(\log\log T)$ regret against the benchmark strategy.

## C Proof of Proposition 1

We start by stating the proposition more formally.

**Proposition 2.** *Assume that there exist a partition of $[1,T]$ into $K$ intervals $I_1,\ldots,I_K$ such that the $i$-th component of the loss vectors within each interval $I_k$ are drawn independently from a fixed probability distribution $\mathcal{D}_{k,i}$ dependent on the index $k$ of the interval and the identity of expert $i$. Furthermore, assume that at any time $t$, there exists a unique expert $i_t^*$ and gap parameter $\delta > 0$ such that $\mathbb{E}\left[\ell_{t,i_t^*}\right] \leq \mathbb{E}\left[\ell_{t,i}\right] - \delta$ holds for all $i \neq i_t^*$. Then, the regret $\mathrm{FTL}(w)$ with parameter $w > 0$ is bounded as*

$$\mathbb{E}\left[\mathfrak{R}_T(\mathrm{FTL}(w),y_{1:T})\right] \leq wK + NT\exp\left(-\frac{w\delta^2}{4}\right),$$

*where the expectation is taken with respect to the distribution of the losses. Setting $w = \left\lceil 4\log(NT/K)/\delta^2 \right\rceil$, the bound becomes*

$$\mathbb{E}\left[\mathfrak{R}_T(\mathrm{FTL}(w),y_{1:T})\right] \leq \frac{4K\log(NT/K)}{\delta^2} + 2K.$$

*Proof.* The proof is based on upper bounding the probabilities $q_t = \mathbb{P}\left[b_t \neq i_t*\right]$ for all $t$. First, observe that the contribution of a round when $b_t = i_t^*$ to the expected regret is zero, thus the expected regret is upper bounded by $\sum_{t=1}^T q_t$. We say that $t$ is in the $w$-interior of the partition if $t \in I_k$ and $t > \min\{I_k\} + w$ hold for some $k$, so that $b_t$ is computed solely based on samples from $\mathcal{D}_k$. Let $\hat{\ell}_t = \sum_{s=t-w-1}^{t-1} \ell_s$ and $\bar{\ell}_t = \mathbb{E}\left[\ell_t\right]$. By Hoeffding's inequality, we have that

$$q_t = \mathbb{P}\left[b_t \neq i_t^*\right] \leq \mathbb{P}\left[\exists i : \hat{\ell}_{t,i_t^*} > \hat{\ell}_{t,i}\right]$$

$$\leq \sum_{i=1}^N \mathbb{P}\left[\left(\bar{\ell}_{t,i} - \bar{\ell}_{t,i_t^*}\right) - \left(\hat{\ell}_{t,i} - \hat{\ell}_{t,i_t^*}\right) > \delta\right]$$

$$\leq N\exp\left(-\frac{w\delta^2}{4}\right)$$

holds for any $t$ in the $w$-interior of the partition. The proof is concluded by observing that there are at most $wK$ rounds ouside the $w$-interval of the partition and using the trivial upper bound on $q_t$ on such rounds. $\qquad\square$