[Reviews · NeurIPS 2014]

Submitted by Assigned_Reviewer_11

Summary:
In this paper the authors present a master algorithm that combines two learning algorithms and allows to exploit easy data while preserve the robustness in hard cases. The first input algorithm is designed for the worst-case (such as OGD), and the second input algorithm (which the authors call 'benchmark') provides good performance in case of easy data (such as FTL), but not necessarily in case of hard data. The master algorithm is able to smoothly mix the predictions of the two algorithms, and adapt to the first algorithm in case of hard data, but to the second in case of easy data. The authors provide regret bound for the master algorithm, comparing to regret of the two input algorithms. Some applications are reported to various learning problems, and finally some empirical results compare the regret of the proposed algorithm to previous algorithms.

Points for and against the paper:
The paper addresses an interesting problem. It is well organized and pleasant to read. The authors fairly compare their algorithm to previous algorithms. I liked the generality of the algorithm, and the applications in section 3 (it could be nice to provide application of (A,B)-PROD to online regression or classification tasks).
However, the theoretical results are not strong enough to my opinion. The good regret in case of easy data comes at a cost of addition factor of sqrt{TlogT} in the worst-case. The bound in Theorem 4 seems to be worse than AOGD bound.

Comments about the empirical results section:
- The plots are very small.
- In the setting 2 plot it is not clear where the blue line (FTL) is.
- Please add a short summary of the difference between the settings (where is easy/hard data).
- From the plots it seems that the (A,B)-PROD algorithm is not the best in any case. It could be nice to find a case where the (A,B)-PROD algorithm provides the best results.
- The discussion about the comparison between (A,B)-PROD and FLIPFLOP in the empirical results seems to be in a contradiction to the discussion in page 5. In settings 2,3,4 where FTL performs well (easy data) the FLIPFLOP outperforms (A,B)-PROD. This is a contradiction to the theory in lines 245-248.

Additional comments:
- The discussion at the top of page 3, where U_t and V_t are used, seems redundant.
- In line 113 – what is F^star ?
- In lines 155-156 – "the output is set to b_t with probability s_t, and a_t otherwise". Maybe it should be a_t with probability s_t ?
- In lines 250-252 – The sentence starting at "The other advantage…" is not clear. In theorem 2 there is a bound on the regret of the algorithm in terms of the regret of ADAHEDGE. So what is the difference between this and de Rooij [8] ?
- In line 395 – "Figure 4" should be "Figure 3".
- In line 513 – the first inequality is redundant.
- It could be better to add the definition of eta_t in line 543 to the description of algorithm 1.
- In line 545 - mu_B and mu_A should be w_{1,B} and w_{1,A} ?
- In line 554 – add parentheses to O..
- In line 586 – "ouside".
- In general it seems that the authors tend to use semicolon where comma should be used.
Summary: The theoretical results are not strong enough and the empirical results section should be improved significantly.

Submitted by Assigned_Reviewer_34

This work presents a new way to exploit 'easy' data in online convex
optimization and in the bandit setting. Given a strategy A that has good
worst-case guarantees, and another opportunistic strategy B that might,
for some data, get much smaller regret, it is shown how to combine A and
B into a single strategy that gets the best of both worlds, at the cost
of adding O(sqrt(T*log(T))) to the regret of A and a small constant to
the regret of B. The same approach is shown to work for prediction with
expert advice, online convex optimization and a relaxation of the bandit
setting in which *two* arms can be observed in every round.

This is pretty cool, because previous authors have needed to use
complicated special-purpose strategies that depended on A and B to
achieve similar results.

Applications are given that 1) take B to be Follow-the-Leader and get
performance similar to the FlipFlop algorithm of De Rooij et al. [8]; 2)
address a recent open problem about tracking the best expert, and 3)
adapt to worst-case and stochastic bandits as in the work of Bubeck and
Slivkins [5]. In these applications, the results of prior work are not
exactly recovered, but the authors generally provide a good discussion
of the differences.

The paper makes me wonder about a natural follow-up question: Is it
possible to generalize the approach to include more than one
opportunistic benchmark strategy B and get small regret with respect to
all of them? Currently the additive term O(sqrt(T*log(T))) prevents
nesting the approach multiple levels.

Some remarks:

Tracking the best expert:
- the proposed method needs to know the gap delta between the best
expert in each interval and the second-best, which is unknown in
practice. I would suggest adding a comment that mentions this.

The comparison to FlipFlop on p.5:
- This comparison focusses on using specific parameters for FlipFlop.
These parameters trade off the performance in the worst-case with
the overhead with respect to FTL. So perhaps a fairer comparison
could be made by setting them to make the worst-case for FlipFlop
and for (A,B)-Prod the same. Alternatively, the discussion might be
simplified by not focussing on particular parameter values too much.
- The comment about competing with the actual regret of
AdaHedge vs its regret bound seems misplaced, because the difference
between the two is negligible compared to the additional
sqrt(T*log(T)) term.

line 267: why does D-Prod guarantee regret of O(T^{1/4}) when FTL and
AdaHedge both have regret Theta(sqrt(T))? Suppose we generate losses
for 2 experts from the uniform distribution (as in the lower bound
from "How to use expert advice"), then any algorithm will have
regret Theta(sqrt{T}), so how can this be true? Which algorithm is
used as the benchmark in this case?

Minor comments:

In figure 2, w_{1,A} is not really a parameter, because it is fixed to
be 1-w_{1,B}.
In Thm 1, why introduce mu_A and mu_B, when they are the same as
w_{1,A} and w_{1,B}?
In Corollary 1, I think your result requires C >= e^4 or something.
line 245: "a factor of" -> "an additive term of" (factors are multiplicative)
Proposition 1: say that there is a proof in the appendix
In the discussion of your result for the bandit setting, you could also
mention the recent paper "One Practical Algorithm for Both Stochastic
and Adversarial Bandits" by Seldin and Slivkins at ICML 2014.
line 488: there is a minus missing: -log(1-eta)/eta is increasing
line 497: doesn't this hold with equality?

Summary: This is pretty cool, because previous authors have needed to use complicated special-purpose strategies to achieve similar results.

Submitted by Assigned_Reviewer_42

A well-known result in the theory of prediction with expert advice states that for bounded convex loss functions the learner can do as well as the best expert up to a regret term of the order O(\sqrt{T}), where T is time (or the length of the sequence).

This result cannot be improved generally. However the O(\sqrt{T}) bound holds in the worst case and does not necessarily makes sense in particular cases; this situation is common in the theory of prediction with expert advice. The O(\sqrt{T}) order of magnitude may be too loose when dealing with easily predictable sequences ("easy data") leading to small loss.

This paper proposes an algorithm (A,B)-PROD that merges two experts (or meta-experts) A and B and at the cost of slightly increasing the regret term for A (namely, going from O(\sqrt{T}) to O(\sqrt{T\log T})) achieves a constant regret for B.

The authors formulate the result in the general framework of convex optimisation (Theorem 1 and Corollary 1) and then apply it to a special case (Theorem 2) of prediction with expert advice where A is AdaHedge and B is FTL (Follow the Leader). There are reasons to believe that FTL should perform well and suffer very small loss in a large class of "simple" situations. The choice of AdaHedge and FTL is also motivated by comparison with an earlier algorithm called FlipFlop; the authors carefully perform the comparison.

However it should be noted that the construction in the paper is not limited to AdaHedge and FTL in the same sense as the earlier PROD algorithm appears to be limited to the average of experts. Any two algorithms can take their place.

Applications to tracking the best expert and bandit problem are considered.

I believe the result of the paper is fundamental for prediction with expert advice and has a wider significance for related areas.
Summary: The paper proposes a merging algorithm for two experts in the convex case achieving O(\sqrt{T\log T})) regret for one expert and constant regret for another. This is a fundamental contribution to prediction with expert advice.
Author Feedback
Author rebuttal: We thank the reviewers for their very helpful and thoughtful feedback. We will revise our paper to address the reviewers' suggestions to improve readability. In the following we address the main questions from the reviewers.

Rev.1
1. "Theoretical results are not strong enough to my opinion."
The result in Corollary 1 is near-optimal (constants apart) in a zero-order worst-case sense, since on difficult data \sqrt{T} is indeed a lower-bound. In that sense we only loose a factor \sqrt{log(T)}. Furthermore notice that the additive term is indeed O(\sqrt{C log C}) which may be significantly smaller than O(\sqrt{T log T}). Whether higher-order bounds (eg, with explicit dependency on L*) can be derived or whether \sqrt{Tlog(T)} is indeed the best we can hope for AB-prod (or similar algorithms) is still an open question.

Also note that Thm. 6 in [9] shows that removing this \sqrt{log(T)} factor is in some sense difficult even in the special case when the benchmark is a uniform average of all experts. This result also indicates that improving our general guarantees is difficult in the same sense. We will clarify this point in our final version.

2. "The bound in Theorem 4 seems to be worse than AOGD bound."
Corollaries 3.1 and 3.2 in [4] show O(\sqrt{T}) for convex and O(log(T)) for strongly convex functions. At the cost of an additional regret of O(\sqrt{Clog C}) in the first case and an additive constant in the second case, the bounds in Thm. 4 match those of AOGD.

3. Empirical results.
The numerical simulations considered in Sect.4 are the same as in [8] and they are intended as a sanity check for the performance of AB-Prod. Apart from setting 1, the others are "easy" data since FTL has constant regret. AB-Prod is indeed not the best in these settings but it performs better than FlipFlop in the first setting and it is always competitive in the others. Notice that in practice parameters should be hand-tuned (at least in terms of constants in the definition of eta) and the ordering between the algorithms may easily change. Notice that the apparent incoherence between the theoretical comparison with FlipFlop in lines 245-248 and the empirical results is due to the fact that we are dealing with upper bounds whose tightness depend on the specific sequence at hand. We have not actively tried to construct a specific example in which AB-Prod is the best but it is possible by looking at the discussion in L245-248. Finally, in general we expect an ad-hoc and carefully tuned algorithm for each of the scenarios we consider (expert advice, convex optimization, and so on) to perform better than our general-purpose meta-algorithm.

4.The sentence "The other advantage…" is not clear.
To see the difference we should refer to Theorem 1 where the bound is given on the total *actual loss* rather than an *upper-bound* on the regret (as for FlipFlop). This means that whenever there is a difference between the actual performance and the upper-bound, AB-Prod may perform better.

Rev. 2
1. More than one opportunistic benchmark.
This is indeed a very interesting question, although it seems that competing with multiple benchmarks is impossible in general. This is easy to show on a simple example with two experts and two benchmarks corresponding to each expert. We will add a discussion in the final version of the paper.

2. Tracking the best expert.
This is correct, the optimal choice of the window depends on the gap delta. The theorem still holds for any arbitrary choice of w (see proof in the supplementary).

3. Choice of the parameters in the experiments.
Performing a fair comparison between algorithms with different parameters is always challenging. Since our numerical simulations are intended to be a sanity check for the theory, we decided to use the parameters obtained by optimizing the bounds for each algorithm.

4. Comment about competing with the actual regret of AdaHedge vs its regret bound seems misplaced.
While we agree that the difference actual regret vs regret bound may be small compared to the additive term, we should also consider that the additional term O(\sqrt{C log C}) may be significantly smaller than O(\sqrt{T log T}) and thus it may not necessarily cancel the advantage in the first term.

5. Why does D-Prod guarantee regret of O(T^{1/4})?
D-Prod provides a direct bound on the loss of the form [Loss(D-Prod) <= Loss(any expert) + Otilde(sqrt{C})] and a bound of the loss wrt to the loss of the benchmark (FTL) plus a constant term. If C is of order O(sqrt(T)) (loss of FTL), then from the first bound we have that the loss of D-Prod is upper bounded by the loss of the best expert plus O(sqrt (sqrt T)). It is possible to construct (deterministic) sequences satisfying these conditions. We will elaborate this point in more detail in the final version of the paper.

6. Reference to "One Practical Algorithm for Both Stochastic and Adversarial Bandits"
This is indeed a great paper that we found after our submission and it seems to effectively deal with the "true" bandit setting for easy and hard data. We are still trying to understand to which extent an AB-Prod approach may be able to solve that setting with similar guarantees.

Rev. 3
Thank you for your positive review!